# An Innovative, Unobtrusive Approach to Investigate Smartphone Interaction in Nonaddicted Subjects Based on Wearable Sensors: A Pilot Study

**DOI:** 10.3390/medicina55020037

**Published:** 2019-02-04

**Authors:** Alessandro Tonacci, Lucia Billeci, Francesco Sansone, Antonella Masci, Anna Paola Pala, Claudio Domenici, Raffaele Conte

**Affiliations:** 1Institute of Clinical Physiology-National Research Council of Italy (IFC-CNR), Via Moruzzi 1, 56124 Pisa, Italy; lucia.billeci@ifc.cnr.it (L.B.); francesco.sansone@ifc.cnr.it (F.S.); pala@ifc.cnr.it (A.P.P.); domenici@ifc.cnr.it (C.D.); raffaele.conte@ifc.cnr.it (R.C.); 2University of Pisa, School of Engineering, Largo Lucio Lazzarino 1, 56122 Pisa, Italy; masci.antonella@yahoo.it

**Keywords:** internet addiction, quality of life, smartphone addiction, social anxiety

## Abstract

*Background and objectives:* Smartphones are playing a pivotal role in everyday life, due to the opportunity they grant in terms of simplifying communication, entertainment, education and many other daily activities. Against such positive characteristics, smartphone interaction can result, in particular cases, in dangerous smartphone addiction patterns, possibly leading to several long-term detrimental psychophysiological conditions. Therefore, this pilot aims at assessing the feasibility of using an innovative approach, based on unobtrusive wearable sensors, used for the first time in this specific topic, and psychological questionnaires, to investigate the links between stress and emotions in a group of young, nonaddicted individuals performing smartphone interaction. *Materials and methods:* 17 volunteers were enrolled for the present study. The study protocol was divided into three phases, with an initial resting state (baseline) of three minutes, a smartphone interaction session (task) of the same length, and a final resting state (recovery), lasting three minutes. In the overall procedure, electrocardiogram (ECG) and galvanic skin response (GSR) measurements, both monitored by wearable sensors, were acquired in order to assess the functioning of the autonomic nervous system (ANS). *Results:* A significant decrease was seen in pNN50 during the smartphone interaction with respect to the baseline (*Z* = −2.675, *p* = 0.007), whereas the Low-to-High Frequency (LF/HF) ratio at task was somewhat correlated with phubbing behaviors (*r* = 0.655, *p* = 0.029), assessed through dedicated questionnaires. *Conclusions:* Taken together with the slight changes in GSR data, such results suggest the feasibility of this approach to characterize the ANS activation during smartphone interaction among young individuals. Further studies should enlarge the study population and involve smartphone-addicted subjects in order to increase the scientific and clinical relevance of such findings.

## 1. Introduction

Nowadays, the employment of technologically advanced mobile devices, including smartphones, represents an essential part of daily life. According to statistics, the number of smartphone users worldwide will exceed 2.5 billion by 2018 [1], with higher penetration among younger people [2]. In Italy alone, 33.3 million smartphone users are estimated by 2018 [3], roughly representing 55% of the overall population, thanks to the various opportunities offered by those devices, spanning communication, following up on appointments and calls, social networking, entertainment, internet access, music, banking and education. The extreme versatility and attractiveness of such devices [4] can give rise to the so-called “smartphone addiction” [5], mainly frequent among young people (see, for example, [6,7]). Smartphone addiction is defined as an “uncontrolled and excessive use of the phone at a level that will affect the daily lives of users” [8,9], possibly leading to a range of negative consequences on daily life. Such effects include physical and psychological problems, such as disorders related to the musculoskeletal system, pain, impairment in daily routines, sleep disorders, decreased physical activity, stress, deprivation, decreased academic performance, and changes in family and social relationships, including loneliness, as well as in communication abilities [10,11,12,13]. A growing number of studies, mainly conducted on young individuals and students in particular, confirm the negative impact of excessive smartphone use on psychological wellbeing [14,15], with evidence also concerning aggression, worry and anger in subjects with problematic smartphone use [16]. In addition, the growing phenomenon of “phubbing”, in which a person snubs another in a social setting by concentrating on their phone instead of having a conversation, highlights the real negative consequences of the lack of communication that detrimentally affects relationships and feelings of personal wellbeing [17].

Taken together, those facts could lead to serious long-term consequences, which could be extremely burdensome for civil society at large (see [18] for some examples).

In smartphone addiction, like in other kinds of dependence, including technological ones, people usually repeat the smartphone interaction behavior leading to addiction. This act usually brings them short-term positive feelings that give them pleasure [6]. On the other side, preventing a person from using their smartphone elicits symptoms of tension, restlessness and deprivation, as occurring, for example, in drug addiction [19,20].

The monitoring of such psychophysiological processes is often difficult to perform, sometimes requiring expensive processes difficult to undertake in real time and causing annoyance to the subject evaluated. The use of minimally obtrusive wearable sensors could reduce this burden, without, at the same time, sacrificing accurate and timely estimation of key psychophysiological parameters.

Therefore, in this work we aimed at: (i) evaluating the feasibility of using wearable sensors for the extraction of autonomic parameters; (ii) estimating the effect of smartphone interaction on such parameters, extracted from electrocardiogram (ECG) and galvanic skin response (GSR) signals, in nonaddicted young individuals; and (iii) assessing eventually existing relationships between smartphone interaction, autonomic parameters and psychological questionnaires describing smartphone/technology addiction, as well as some features related to social behavior.

## 2. Materials and Methods

### 2.1. Study Population

17 healthy volunteers (7 males, 10 females, aged 38.5 ± 7.5 years) were enrolled for the present study. All subjects gave their informed consent for inclusion before they participated in the study. The study was conducted in accordance with the Declaration of Helsinki, and the protocol was approved by the internal institutional Ethics Committee of CNR-IFC in 2017. Inclusion criteria were a minimum education level of 13 years (secondary school completed), possession of an Android-or iOS-based smartphone with enabled internet connection, and good reported ability in interacting with their own smartphone. Exclusion criteria included the presence of associated neurological conditions, sensory (visual) impairment reported, or inability/unwillingness to sign informed consent.

All the subjects were asked to undergo physiological signal measurements at rest and during a smartphone interaction, and to compile a battery of psychological questionnaires at home, in order to reduce the burden associated with the present protocol.

### 2.2. Signal Acquisition

Participants were equipped with devices for the acquisition of physiological signals, including ECG and GSR.

Both signals were acquired with unobtrusive wearable sensors realized by Shimmer Sensing, Inc. (Dublin, Ireland), and more specifically the single-lead Shimmer™ ECG Unit and the Shimmer3™ GSR+ Unit according to a protocol previously developed [21,22]. Both devices were connected by Bluetooth to a tablet, running a proprietary graphical user interface.

ECG and GSR signals were acquired during three different phases, detailed below:baseline (3 min): basal measurement. The subject, seated in a comfortable chair, was asked to relax during this phase;task (3 min): The subject was asked to freely interact (chatting, sending messages, using apps) with their own smartphone during this phase. All the subjects visited social networks (Facebook, Instagram and Twitter) during the acquisition, without directly messaging with other users;recovery (3 min): post-task basal measurement. The subject was asked to relax during this phase, similarly to the baseline.

### 2.3. Psychological Questionnaires

A battery of validated psychological questionnaires was sent to the volunteers enrolled for compiling at home, in order to reduce the burden of the protocol, the questionnaires being particularly long. The battery included:the 10-item Self-Scoring Self-Control Scale (SCS): self-assessment scale for self-control evaluation [23];10-item version of the Fear of Missing Out scale (FoMO): self-assessment scale related to the social exclusion phobia [24];Internet Addiction Test (IAT), Italian Version: self-assessment scale for the evaluation of internet-related addiction [25];Phubbing Questionnaire (PQ): self-assessment scale for evaluating “phubbing” behaviors. PQ produces three different subscores, including “Phubbing” (PB), “Being Phubbed” (BP), and the overall “Total Phubbing” (TP), sum of the two previous ones [26];Smartphone Addiction Scale (SAS): self-assessment scale for evaluating the smartphone-related addiction [6].

### 2.4. Signal Analysis

#### 2.4.1. ECG

ECG signal was analyzed through a Matlab (Mathworks, Natick, MA, USA)-based interface properly developed, which allows one to calculate the associated tachogram according to the Pan-Tompkins algorithm [27] and to extract both time- and frequency-domain features associated with the signal [28,29,30].

More specifically, extracted features include:
in the time domain:
○heart rate (HR): number of contractions of the heart occurring per time unit, expressed in bpm. The HR is normally related to sympathetic activation of the autonomic nervous system;○standard deviation of normal-to-normal R–R intervals (SDNN): measure of heart rate variability (HRV), expressed in ms. SDNN is normally affected by both sympathetic and parasympathetic components [31];○changes in successive normal sinus (NN) intervals exceeding 50 ms (pNN50), expressed as a percentage. Like other HRV measures, pNN50 also indicates the overall activity of the autonomic nervous system; however, pNN50 is often considered as a reliable indicator of the parasympathetic activity.in the frequency domain:
○normalized component of the power spectral density of the ECG signal at low frequency (0.04–0.15 Hz) (nLF). nLF is related to both sympathetic and parasympathetic activity;○normalized component of the power spectral density of the ECG spectrum at high frequency (0.15–0.4 Hz) (nHF). nHF is related to parasympathetic activity [28];○low- vs high-frequency component of the power spectral density of the ECG spectrum (Low-to-High Frequency (LF/HF) ratio). LF/HF ratio is considered as a sort of balance between sympathetic and parasympathetic activity.

Frequency-domain parameters were extracted by the power spectral density estimated by the Welch method [32].

#### 2.4.2. GSR

GSR signal was analyzed through the Matlab-based software Ledalab V3.4.9 (General Public License (GNU)) [33]. With the help of this tool, several characteristic features were extracted for each phase, including the overall mean GSR signal and its tonic component. The phasic component was extracted, however, it was not taken into account in this work due to the rationale of the study, which aimed at comparing the signal in the various phases and not its response to a given stimulation (Figure 1).

### 2.5. Statistical Analysis

Statistical analysis was performed with the SPSS v.23 (IBM Corporation, Armonk, NY, USA) tool.

The first step aimed to estimate the distribution of the variables investigated through statistical methods for the assessment of normality, including the Shapiro-Wilk Test [34]. This analysis demonstrated the non-normal distribution of all the scale variables studied.

Therefore, successive characterizations were performed by nonparametrical testing, including variance analysis with a two-way Friedman’s test, and post-hoc analysis, in case of significance, with the Wilcoxon signed rank test for the comparison of physiological signals between the three phases. A two-tailed Spearman’s test was employed for correlation analysis.

## 3. Results

### 3.1. ECG Parameters

A significant difference was observed among the three phases described above in the pNN50 (*F* = 10.941, *p* = 0.004), with a significant decrease reported during the task with respect to baseline (*Z* = −2.675, *p* = 0.007), while no variations were seen between task and recovery data (Figure 2).

We failed to find significant differences for all the other ECG-related parameters, including heart rate (*F* = 0.471, *p* = 0.790), SDNN (*F* = 3.294, *p* = 0.193), normalized low frequency (*F* = 3.970, *p* = 0.137), normalized high frequency (*F* = 3.970, *p* = 0.137), and LF/HF ratio (*F* = 3.970, *p* = 0.137).

### 3.2. GSR Parameters

Both mean GSR (*F* = 17.714, *p* < 0.001, Figure 3) and tonic GSR (*F* = 16.714, *p* < 0.001) signals were significantly different between the three phases. Going into depth in this analysis, both mean GSR (*Z* = 2.668, *p* = 0.008) and tonic GSR (*Z* = 2.480, *p* = 0.013) were higher at task with respect to the baseline signal, however, both reached their respective peaks during the recovery, which displayed higher values with respect to the task (*Z* = 2.856, *p* = 0.004 for mean GSR, *Z* = 2.794, *p* = 0.005 for tonic GSR).

### 3.3. Questionnaires

After multiple comparison correction, a significant negative correlation was seen between Self-Control Scale and Internet Addiction Test scores (*r* = −0.679, *p* = 0.022), as well as between Self-Control Scale and Phubbing Score (*r* = −0.618, *p* = 0.043). No other correlations were retrieved.

### 3.4. Correlations between Physiological Signals and Questionnaires

Significant correlations (Figure 4) were reported between the LF/HF ratio acquired at task and the Phubbing Score (*r* = 0.655, *p* = 0.029), between the ΔLF/HF ratio (task-baseline) and the same Phubbing Score (*r* = 0.655, *p* = 0.029), and between the ΔLF/HF ratio (recovery-task) and the Total Phubbing Score (*r* = −0.769, *p* = 0.006). On the other hand, no significant correlations were found between GSR parameters and psychological questionnaires.

## 4. Discussion

Smartphone interaction does appear to bring to slight changes in some well-defined autonomic parameters extracted from ECG and GSR signals.

Specifically, the ECG-based pNN50, which measures the overall activity of the ANS and was proposed as a measure of the parasympathetic activity [35], decreased from baseline to task. Therefore, young nonaddicted individuals decrease their parasympathetic activity during smartphone interaction. Such a decrease does not appear to be reversible (or maybe its eventual reversibility could be masked by the relatively low time frame for the recovery phase), since recovery-phase results are comparable to task data, and this fact could eventually mask other factors, including the possible annoyance brought by the examination to each volunteer. Indeed, the increased GSR signal-which normally follows different dynamics from the ECG-during recovery with respect to task, in turn higher than in the baseline, probably suggests such an annoyance effect concerned with the development of the examination. Considering the involvement of only nonaddicted individuals in this study, it is reasonable that the magnitude of the effect caused by the smartphone interaction (here, the evidence appears to be towards a lower parasympathetic activity during the task) is quite negligible, as evidenced by the reported increase in the GSR signal at recovery with respect to the task. It could be speculated that in a population composed by addicted individuals, such effects would be more evident and possibly highlighted by variations in this specific signal.

Indeed, variations in the LF/HF parameter during the task with respect to the baseline, a good indicator for sympathetic vs parasympathetic activation within the ANS, were seen to be correlated with the Phubbing score, while the LF/HF variation at recovery with respect to the task was inversely correlated with the Total Phubbing score. Taken together, such results could indicate that people, even when not addicted to smartphones, are more emotionally activated when interacting with such a device. This concept is even more evident for those subjects that are more subject to phubbing behaviors, in turn related to abnormal sociality [36,37]. This is coherent with the observations of Matar Boumosleh and Jaalouk [38], who display an association between anxiety/depression and stress in general addiction and smartphone addiction. A similar outcome was also found by Lee and colleagues [39], confirming the likelihood of this association and the effect of social interaction anxiety on compulsive smartphone use, even in the general adult population [40].

To the best of our knowledge, this is the first study using psychophysiological correlates for investigating the acute effects of smartphone interaction on the end-users. Recent works have focused on smartphone and technology addiction assessed through existing questionnaires, finding sociality-related problems [41] and inability in concentrating during study sessions for more addicted subjects [42]; however, their influence on physiological signals is still a dark side in actual research, making the comparison with existing literature impossible.

Our research could therefore be an initial milestone into the investigation of possible methods to assess smartphone addiction even in apparently nonaddicted subjects. This fact could represent a useful diagnostic aid in future, considering that an early intervention-with mood regulation treatments, for example-in addicted subjects could have a positive effect both on the mood and on smartphone use, as demonstrated in [43], highlighting the importance of studying new methods to find early, predictive cues for smartphone addiction in apparently nonaddicted subjects.

### Limitations

The results obtained in this study should be taken into account in light of some limitations. At first, the pilot nature of this study, performed on a rather limited number of subjects, makes it impossible to draw generalizable conclusions about the effective relationship between psychophysiological signals and smartphone addiction, investigated through questionnaires. Therefore, future research should address this limit involving a larger number of volunteers in the analysis.

Secondly, the present work included data of nonaddicted subjects only. However, the most scientifically and clinically relevant retrievals should be looked for among addicted individuals, and future research will deal with this point.

Finally, the cross-sectional nature of this work is appropriate for the present investigation, but should be overcome in future studies, in order to assess causality of the relationship between stress/anxiety, smartphone addiction and related behavior.

## 5. Conclusions

Summarizing, the present work demonstrated the feasibility of this approach to characterize the ANS activation in response to smartphone interaction in a cohort of young, nonaddicted volunteers. Even in a relatively small cohort, a slight effect of this interaction in the sympathetic/parasympathetic overall balance was seen, with an increase of the sympathetic contribution during the task administration, somewhat correlated with some well-defined psychological characteristics connected to the technological addiction.

Future works could take advantage of the positive outcomes obtained by the present research, for example, by enlarging the study population to smartphone-addicted subjects, with the involvement of a larger cohort in order to obtain more statistically relevant results.

In addition, the use of further technological tools to extract physiological signals including, for example, wearable wireless systems for the acquisition of electroencephalography (EEG), could allow us to investigate the problem in more depth, without sacrificing, at the same time, the noninvasiveness of the proposed approach.

## Figures and Tables

**Figure 1 medicina-55-00037-f001:**
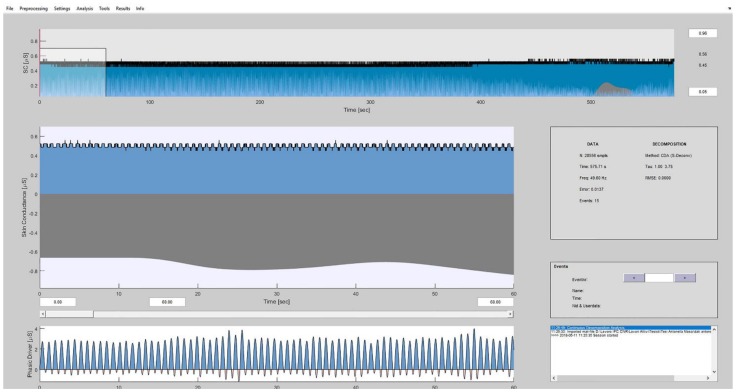
Matlab-based Ledalab interface for the analysis of the galvanic skin response (GSR) signal.

**Figure 2 medicina-55-00037-f002:**
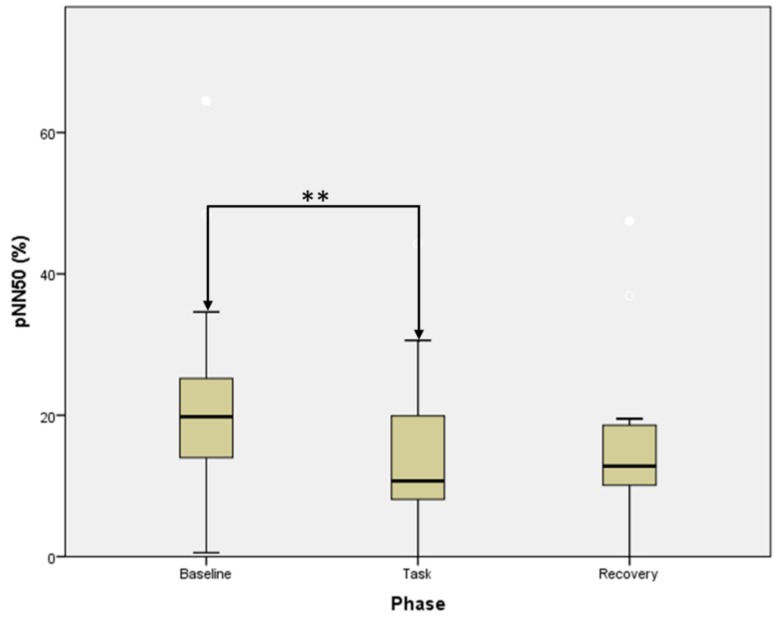
pNN50 trend over the test phases (** statistical significance at *p* < 0.01 according to the Wilcoxon signed rank test). Error bars indicate the range of distribution; the box, the interquartile range; the horizontal line, median value.

**Figure 3 medicina-55-00037-f003:**
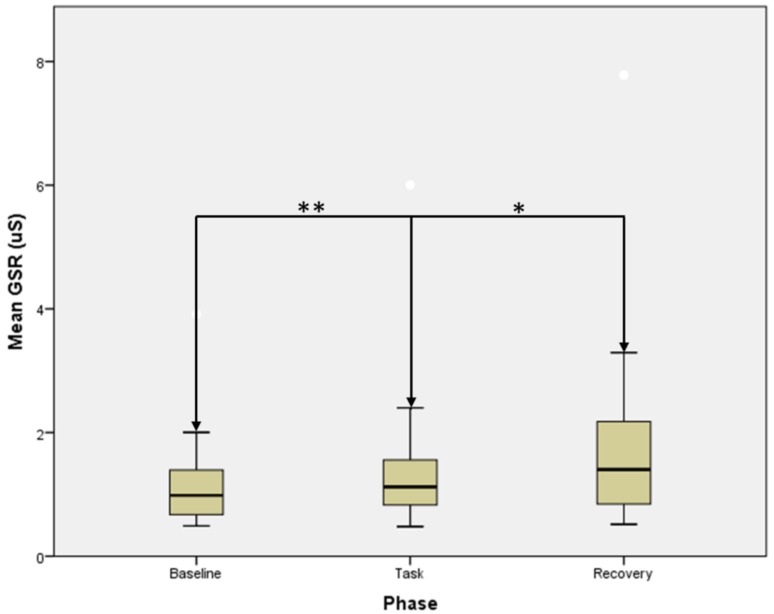
Mean GSR trend over the test phases (* statistical significance at *p* < 0.05; ** statistical significance at *p* < 0.01 according to the Wilcoxon signed rank test). Error bars indicate the range of distribution; the box, the interquartile range; the horizontal line, median value.

**Figure 4 medicina-55-00037-f004:**
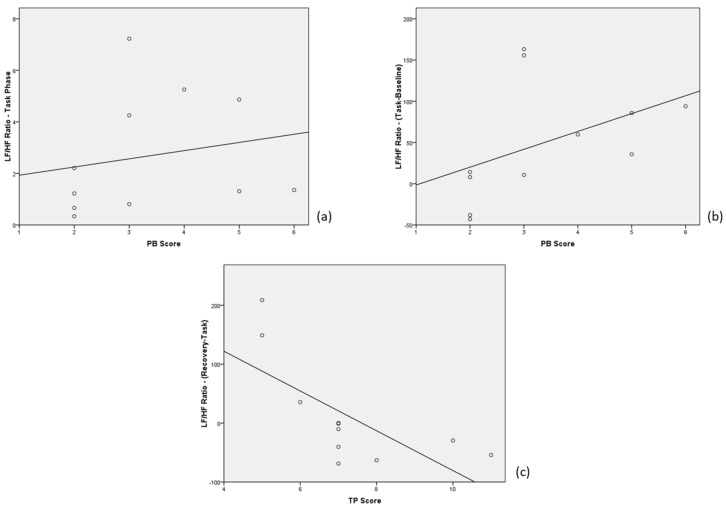
Significant correlations between physiological signals and questionnaires: (**a**) LF/HF (Low-to-High Frequency) ratio at task vs Phubbing Score (PB); (**b**) ΔLF/HF ratio (task-baseline) vs Phubbing Score (PB); (**c**) ΔLF/HF ratio (recovery-task) vs Total Phubbing Score (TP).

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
