# Peer review of "An Innovative, Unobtrusive Approach to Investigate Smartphone Interaction in Nonaddicted Subjects Based on Wearable Sensors: A Pilot Study"

_medicina, 2019, doi:10.3390/medicina55020037_

Round 1

Reviewer 1 Report

This is an interesting pilot study interested in determining if there are physiological changes that underlie the use of smartphones in people. I think the introduction well positions the paper given that people do report more positive experiences/interactions with their smartphones which might likely have physiological effects as well.

One of the issues with the methods is that the length of time tested may not have been long enough to see significant changes (3 minutes?). Also did the authors keep track of what the participants did on their phones? My specific comments are below:

- under section 2.2: can the authors please clarify if "3'" = 3 minutes? Please specify more explicitly.

- under section 2.2: do you have more information on what the participants were doing when they were allowed to freely interact? potentially there might be an effect if they were just browsing vs. interacting with others using chat/messaging apps

- under section 2.3: I'm confused as to why the Autism scale was used (AQ). This doesn't seem to fit with the intro/hypotheses? does it need to be in the paper?

- under section 2.4: I would attempt to be a little more descriptive here; for example, what can HR and HRV give us information on? the physiological arousal/sympathetic/parasympathetic nervous system activation; why is this important? emotional experiences with smartphones are likely to be related to physiological experiences

- under section 2.5: the authors mention that they test for normal distributions, but don't report if any of the variables were non-normal , etc.

- results section: my only suggestion is to reduce the number of abbreviations or state more explicitly what these correlations mean as the short forms become too difficult to cross-reference.

- under section 3.3: you could describe the correlation "a significant negative correlation"...

- under 3.4: describe the self-report measures in words, rather than abbreviations; it will make it easier for the reader.

- discussion: "such decrease does not appear to be reversible" - I think this is an overstatement that needs to be corrected. We don't know if it will be reversible given that the study only tested the effects for 3 minutes at a time. There could be a delay in the physiological changes that was not captured because of the design of the study. Also there must be a reversing at some point or else why would the baseline differ.

- line 205-206 - I think you can just delete this sentence because you report similar statements in the paragraph above.

- I think the results need replication; correlations between self-report measures and physiological indices were exploratory and not hypothesis driven. Why might these relationships be significant? It seems unusual that only "Phubbing" is related to the physiological measures. Then again I think you are right in that you only had normal controls and they may not have differed much on the measures of phone addiction. Definitely, more research is needed.

- in the discussion you might want to highlight or circle back to the emotional piece here: smartphones have become comforting to people; do the physiological results represent or show this? the positive emotions induced by smartphone use may then impact physiological changes which are reinforcing.

Author Response

Reviewer-1

This is an interesting pilot study interested in determining if there are physiological changes that underlie the use of smartphones in people. I think the introduction well positions the paper given that people do report more positive experiences/interactions with their smartphones which might likely have physiological effects as well.

Thank you for your kind statement.

One of the issues with the methods is that the length of time tested may not have been long enough to see significant changes (3 minutes?). Also did the authors keep track of what the participants did on their phones? My specific comments are below:

- under section 2.2: can the authors please clarify if "3'" = 3 minutes? Please specify more explicitly.

Thank You. Yes, the time frame was 3 minutes, as we specified in the revised version of the paper. The choice in favor of this frame was performed to keep the time of the assessment reasonably low to not create annoyance to the patient and, at the same time, to collect statistically reliable information from the extraction of physiological parameters (especially those related to the HRV)

- under section 2.2: do you have more information on what the participants were doing when they were allowed to freely interact? potentially there might be an effect if they were just browsing vs. interacting with others using chat/messaging apps

Thank You. Yes, we collected this information. All the subjects involved visited social networks (Facebook, Instagram and Twitter) without interacting by messaging with other users. We tried to add this information in the revised paper.

- under section 2.3: I'm confused as to why the Autism scale was used (AQ). This doesn't seem to fit with the intro/hypotheses? does it need to be in the paper?

Thank You. We added this assessment to evaluate whether social functioning, which is indirectly monitored also by the AQ, could be correlated with a different behavior during the interaction. However, we agree that this eventual hypothesis could be quite difficult to be fully explained. Therefore, we agree with you and deleted this scale from the paper description.

- under section 2.4: I would attempt to be a little more descriptive here; for example, what can HR and HRV give us information on? the physiological arousal/sympathetic/parasympathetic nervous system activation; why is this important? emotional experiences with smartphones are likely to be related to physiological experiences

Thank You. We tried to briefly explain point-by-point the physiological meaning of the different ECG features, according to the work by Shaffer and Ginsberg, 2017, without altering the overall balance of the section.

- under section 2.5: the authors mention that they test for normal distributions, but don't report if any of the variables were non-normal , etc.

Thank You. Our mistake: we corrected accordingly.

- results section: my only suggestion is to reduce the number of abbreviations or state more explicitly what these correlations mean as the short forms become too difficult to cross-reference.

Thank You. We modified accordingly, whenever possible.

- under section 3.3: you could describe the correlation "a significant negative correlation"...

Thank You. We added “negative” as suggested.

- under 3.4: describe the self-report measures in words, rather than abbreviations; it will make it easier for the reader.

Thank You. As above, we modified accordingly, whenever possible.

- discussion: "such decrease does not appear to be reversible" - I think this is an overstatement that needs to be corrected. We don't know if it will be reversible given that the study only tested the effects for 3 minutes at a time. There could be a delay in the physiological changes that was not captured because of the design of the study. Also there must be a reversing at some point or else why would the baseline differ.

Thank You for Your observation. We tried to add a short sentence saying this concept, as “or maybe its eventual reversibility could be masked by the relatively low time frame for the Recovery phase”.

- line 205-206 - I think you can just delete this sentence because you report similar statements in the paragraph above.

Thank You. We deleted the sentence.

- I think the results need replication; correlations between self-report measures and physiological indices were exploratory and not hypothesis driven. Why might these relationships be significant? It seems unusual that only "Phubbing" is related to the physiological measures. Then again I think you are right in that you only had normal controls and they may not have differed much on the measures of phone addiction. Definitely, more research is needed.

I agree with your point. In fact, this study is meant as a pilot, therefore providing initial cues for several future developments, which could definitely arise, giving the importance of the topic and the spread of smartphone diffusion (and, unluckily, addiction) in the general (particularly young) population.

- in the discussion you might want to highlight or circle back to the emotional piece here: smartphones have become comforting to people; do the physiological results represent or show this? the positive emotions induced by smartphone use may then impact physiological changes which are reinforcing.

Thank You. We tried to briefly explain the relationship with stress response in the Discussion section (“[…] Taken together, such results could indicate that people, even when non-addicted to smartphones, are more emotionally activated when interacting with such a device. This concept is even more evident for those subjects which are more subject to phubbing behaviors, in turn related to abnormal sociality [33,34].”).

Reviewer 2 Report

This study is to explore in order to investigate the links between stress and emotions in a group of 17 young, non-addicted individuals performing smartphone interaction. This pilot study aims at assessing the feasibility of using an innovative method, based on wearable sensors and questionnaires. Study design and methods are valid. Results are meaningful. Followings are some comments to improve the quality of the study.

- Since this study uses non-addicted individuals, title is not matched with overall study goal and results.  Revise the title properly. 

- This study argues that it uses an innovative approach. Since ECG is very commonly performed to detect any cardiac problems and GSR is the measure of the continuous variations in the electrical characteristics of the skin. Thus, it may not an innovative approach. Justify it. 

- In introduction section, provide some rigorous literature reviews. 

- Figures 2 and 3 are unnecessary big. Make them a half.

- Exploring the relationship between stress and emotions is very difficult by using an experimental pilot study. Justify it.  

Author Response

Reviewer-2

This study is to explore in order to investigate the links between stress and emotions in a group of 17 young, non-addicted individuals performing smartphone interaction. This pilot study aims at assessing the feasibility of using an innovative method, based on wearable sensors and questionnaires. Study design and methods are valid. Results are meaningful. Followings are some comments to improve the quality of the study.

Thank You for Your kind statement.

- Since this study uses non-addicted individuals, title is not matched with overall study goal and results.  Revise the title properly.

Thank You. We modified the title into “An innovative, unobtrusive approach to investigate smartphone interaction in non-addicted subjects based on wearable sensors: a pilot study”.

- This study argues that it uses an innovative approach. Since ECG is very commonly performed to detect any cardiac problems and GSR is the measure of the continuous variations in the electrical characteristics of the skin. Thus, it may not an innovative approach. Justify it.

Thank You. Your statement is correct; however, ECG and GSR are here used for the first time in the attempt to assess physiological reactions linked with emotions produced by smartphone interaction, thus representing an innovation for this specific field. We tried to add this in the Abstract as “Therefore, this pilot aims at assessing the feasibility of using an innovative approach, based on unobtrusive wearable sensors, used for the first time in this specific topic, and psychological questionnaires, […]”.

- In introduction section, provide some rigorous literature reviews.

Thank You for Your comment. This is a quite novel topic and, since our first submission, some interesting works have been published. We included some relevant literature in the Introduction, with three more works recently published that were added (references 14, 15, 16).

- Figures 2 and 3 are unnecessary big. Make them a half.

Thank You. We significantly reduced the size of the two figures mentioned.

- Exploring the relationship between stress and emotions is very difficult by using an experimental pilot study. Justify it. 

We absolutely agree with you. Indeed, starting from the Abstract, we highlighted the importance of this pilot to verify the feasibility of using this approach to verify the physiological reactions brought by the smartphone interaction. Some indications for future studies, involving a large number of subjects, a number of addicted individuals, and using a different, more comprehensive methodology, are also provided within the text, particularly in the Conclusions section.